# Adaptive Time-Stepping Schedules for Diffusion Models

Yuzhu Chen[1]        Fengxiang He[2]        Shi Fu[1]        Xinmei Tian[1]        Dacheng Tao[3]

[1]Department of Electronic Engineering and Information Science, University of Science and Technology of China, China
[2]School of Informatics, University of Edinburgh, Scotland
[3]College of Computing & Data Science, Nanyang Technological University, Singapore

## Abstract

This paper studies how to tune the stepping schedule in diffusion models, which is mostly fixed in current practice, lacking theoretical foundations and assurance of optimal performance at the chosen discretization points. In this paper, we advocate the use of adaptive time-stepping schedules and design two algorithms with an optimized sampling error bound $EB$: (1) for continuous diffusion, we treat $EB$ as the loss function to discretization points and run gradient descent to adjust them; and (2) for discrete diffusion, we propose a greedy algorithm that adjusts only one discretization point to its best position in each iteration. We conducted extensive experiments that show (1) improved generation ability in well-trained models, and (2) premature though usable generation ability in under-trained models. The code is available at https://github.com/cyzkrau/AdaptiveSchedules.

## 1   INTRODUCTION

Generative modeling stands as a pivotal task in machine learning, with its objective being to acquire knowledge of a probability distribution from available data and produce data samples derived from this acquired distribution. As a subset of generative models, diffusion models excel in achieving state-of-the-art performance for data generation. To date, diffusion models have been applied to a wide variety of generative modeling tasks, such as image generation [Sohl-Dickstein et al., 2015, Ho et al., 2020, Song and Ermon, 2019, Song et al., 2020b, Dhariwal and Nichol, 2021, Nichol and Dhariwal, 2021], text generation [Li et al., 2022, Hoogeboom et al., 2021], text to speech synthesis [Jeong et al., 2021, Kong et al., 2020, Rouard and Hadjeres, 2021] and graph generation [Niu et al., 2020, Huang et al., 2022].

Diffusion models generate data by smoothly transition-

ing from a real-world distribution to noise and back again through forward and reverse processes. Generating samples needs to accurately reconstruct the original data from noise, which is tackled by using neural networks to learn score function and discretization points to simulate the reverse process. Critical to this reconstruction is the time-stepping schedule, which sets the position of each discretization point. This schedule plays a crucial role in determining the quality of the generated data, making it a vital component in the model's performance Chen [2023].

Traditionally, this schedule has been fixed, following a predetermined sequence of steps. However, it does not adapt to the embedding to time or complexity of distributions alone the forward process. This rigidity can lead to inefficiencies, as the model may not utilize the most effective discretization points for a given task, potentially compromising the quality of the generated outputs. Furthermore, these traditional schedules lack theoretical support, which can make it challenging to ensure optimal performance. While methods like Song et al. [2020a], Zhang et al. [2023b] enhance efficiency and quality by selecting a few discretization points at the generating stage, their points accelerate the generating process, rather than selecting effective points and dropping weak points. Besides, their methods of selecting discretization points are still fixed, which means the rigidity still exists, and only decreasing the number of steps should not improve the generating process theoretically.

On the theoretical side, some works provide polynomial bounds of diffusion models' sampling error and provide results about how much discretization points ensure convergence [Chen et al., 2022, Lee et al., 2022, Li et al., 2023, Benton et al., 2023, Lee et al., 2023, Chen et al., 2023]. Their convergence bounds can commonly be separated into two main parts: score estimation error and time discretization error. Score estimation error refers to the difference between score functions and learned score functions, while time discretization error refers to the approximation error of the time-discrete SDE runner. These studies, despite considering various stepping schedules in their assumptions,

often fall short of offering concrete guidance on setting discretization points effectively.

In this paper, we focus on the time-stepping schedule that adapts to the generating process by minimizing a series of convergence bounds. Our contributions are listed as follows.

1. We propose to treat a series of convergence bounds $EB$ as the loss function for optimizing the position of discretization points. It is derived through Girsanov's theorem, and minor terms are treated as hyperparameters. $EB$ contains one term punishing weak score estimator being selected and another term avoiding stepping size from being too high, which means $EB$ considers the balance between score estimator error and time discretization error.

2. We propose to minimize $EB$ for **continuous-time diffusion** by gradient-descent adjusting (GA) discretization points. It works because the score estimator is available at all times within the range $t \in [0, T]$. This technique treats $EB$ as the loss function which not only theoretically narrows $EB$ but also empirically enhances generation quality.

3. We propose to minimize $EB$ for **discrete-time diffusion** by greedily choosing (GC) discretization points. It works because time is embedded only at a finite set of discretization points in discrete-time diffusion, and the score estimator is only available at these points. This approach aims to greedily select every discretization point to its best position, methodically adjusting a single point to its optimal position while freezing others during each choosing iteration.

Empirically, we conduct a Gaussian-target task with a shaking score matching error and show the weakness of fixed time-stepping schedules. After a few iterations of GA, the sampling error decreases and the above weakness is solved, which demonstrates that optimizing $EB$ achieves the purpose of minimizing the sampling error. Furthermore, we tested GC on three popular datasets: MNIST, CIFAR-10, and CELEBA. The results show that GC enhances the image generation quality of diffusion models, regardless of whether they are fully trained or not.

## 2 RELATED WORK

In this section, we provide an overview of the existing literature and research related to our study, including works on stepping schedules, convergence bounds, and training guarantees for diffusion models.

**Stepping schedule** At the beginning of stage diffusion models, linear schedule Ho et al. [2020] and cosine schedule Nichol and Dhariwal [2021] are proposed. To improve the generating speed of diffusion models, DDIM is designed

by Song et al. [2020a], where seldom discretization points are selected for generating. As follows, other innovative methods have been introduced to further refine sampling for diffusion models, such as reverse SDEs Song et al. [2020c] with unique coefficients, skip-step sampling Wang and Li [2024], probability flow ODEs Liu et al. [2022a]. On another line of research, some papers focus on the performance of stepping schedules. Lin et al. [2023] suggests the last step should be started and have zero signal-to-noise ratio to avoid being flawed, while Chen [2023] studies the importance of stepping schedule.

We also note that there are current works focused on finding better stepping schedules for diffusion models. Liu et al. [2023] propose to use a predictor to predict the generation quality from a given stepping schedule, which is trained by the performance of known stepping schedules. This approach works because target distributions and models are similar between situations in training predictors and their usage. Zhang et al. [2023a] propose to maximize a reward function in the generating process to select discretization points. This reward function requires another model to evaluate the quality of generated samples. Different from them, our methods do not rely on the similarity of target distribution or other evaluation models and are theory-supported.

**Convergence bounds** Initial theoretical research into diffusion models lacked precision or faced issues with high-dimensional data, as highlighted by references De Bortoli et al. [2021], Liu et al. [2022b], Pidstrigach [2022] and issues like exponential dependencies in convergence Block et al. [2020], De Bortoli [2022]. Lee et al. [2022], Chen et al. [2022], Lee et al. [2023] improved upon this and offers polynomial convergence guarantees for samplers with L2-accurate score estimates. It is worth mentioning that our considered series of convergence bound $EB$ is inspired by Chen et al. [2022] where a framework of Girsanov's theorem is provided. Following this framework, convergence bounds can be separated into score estimation error and time discretization error. As follows, Chen et al. [2023] provided bounds without Lipschitz score setting and replaced it by a reasonable early stopping setting.

**Training guarantees** To evaluate the approximation error of score estimators, there are some works providing training guarantees for diffusion models. In the paper of Block et al. [2020], DAE loss (used as loss function in the training stage) is proved to be an upper bound of $L^2$ score matching loss, which is crucial to this paper. They also provided $O(\sigma_t^{-4})$ guarantees on score matching loss. As follows, Oko et al. [2023], Gupta et al. [2023] consider the case of neural network with ReLU activation, and provide an $O(\sigma_t^{-2})$ guarantee for score matching loss, which is another foundation of this work.

## 3 NOTIONS AND PRELIMINARIES

In this section, we give an overview of the notions and preliminaries. Through this paper, we study the denoising diffusion probabilistic models (DDPMs) [Ho et al., 2020] where the target distribution $P = P_0$ (with density function $p_0(x)$) is on $\mathbb{R}^d$.

**Forward process** The forward process is an Ornstein-Uhlenbeck process which adds noise to samples from $P$ and is defined by the following equation Santos and Lin [2023].

$$dX_t = -X_t dt + \sqrt{2} dB_t, \ X_0 \sim P_0, \ 0 \leq t \leq T.$$

Here, $\{B_t\}_{t \in [0,T]}$ denotes the $d$-dimensional Brownian motion. For the process $\{X_t\}_{t \in [0,T]}$ alone the forward SDE, we denote $P_t$ the distribution of $X_t$ and denote $p_t(x)$ as its density function. According to the properties of the Ornstein-Uhlenbeck process, $p_t$ has the following analytical form:

$$p_t = p_{(e^{-t})} * g_{(\sqrt{1-e^{-2t}})},$$

where $p_{(e^{-t})}$ refers to the density function of $e^{-t}X_0$, $X_0 \sim P_0$ and $g_{(\sqrt{1-e^{-2t}})}$ denotes the density function of $\mathcal{N}(O_d, (1 - e^{-2t})I_d)$. Besides, $P_t$ converges to $\mathcal{N}(O, I)$ exponentially fast $t$ when $t \to \infty$ in the KL divergence, total variation metric and 2-Wasserstein metric [Bakry et al., 2014, Villani, 2021].

**Backward process** To reverse the forward process, it has been proved in [Anderson, 1982] that for $Y_0 \sim p_T$ and $0 \leq t \leq T$, running the following backward process generates $\{Y_t\}_{t=0}^T$ satisfies $Y_t \sim p_{T-t}$.

$$dY_t = [Y_t + 2\nabla \ln p_{T-t}(Y_t)] dt + \sqrt{2} d\tilde{B}_t. \quad (1)$$

Here, $\{\tilde{B}_t\}$ is another $d$-dimensional Brownian motion that is independent to $\{B_t\}$. Running the backward process needs an unknown term $\nabla \ln p_t$, which is defined as the score function and is always approximated by networks $s_t$ in applications.

**Score matching** To let $s_t$ become a better approximation of the score function, a common method is to train a network with denoising auto-encoder (DAE) loss. For $x$ sampled from $P$ and $z$ sampled from $\mathcal{N}(O_d, I_d)$, DAE loss is defined by:

$$\mathcal{D}_{AE}(s_t, P, t) = \mathop{\mathbb{E}}_{x \sim P, z \sim G} \left\| s_t \left( \sqrt{\alpha_t} x + \sigma_t z \right) - \frac{-z}{\sigma_t} \right\|^2, \quad (2)$$

where $\sqrt{\alpha_t} = e^{-t}$, $\sigma_t = \sqrt{1 - e^{-2t}}$ and $G = \mathcal{N}(O_d, I_d)$. To measure whether $s_t$ is well-trained, a common metric is the $L^2(p_t)$ score matching loss, which is widely used in theoretical analysis of SGMs and is defined by the following equation.

$$\mathcal{D}_{SM}(s_t, p_t) = \mathbb{E}_{x \sim p_t} \| s_t(x) - \nabla \ln p_t(x) \|^2. \quad (3)$$

It is proved in [Block et al., 2020] that the difference between $\mathcal{D}_{AE}$ and $\mathcal{D}_{SM}$ is constant to $s_t$, thus optimizing the denoising auto-encoder loss is equivalent to optimizing the score matching loss.

**Time discretization** Except for the approximation of the score function, running the backward process also needs a simulation for backward SDE. A widely used method is taking many time steps and freezing the value of the coefficient in the SDE at each time step. In details, let initiation be $Z_0 \sim Q_0 = \mathcal{N}(O, I)$ instead of $P_T$, and let $0 = t_0 < t_1 < \cdots < t_N = T - \delta$ be the discretization points ($\delta = 0$ for general setting and $\delta > 0$ for early stopping setting) and denote $h_k = t_k - t_{k-1}$ as the stepping size. For every $k$ and $t \in [t_k, t_{k+1}]$, replace $\nabla \ln p_{T-t}(Z_t)$ as $s_{T-t_k}(Z_{t_k})$ then $Z_t$ satisfies

$$Z_{t_{k+1}} = Z_{t_k} + h_{k+1} [Z_{t_k} + 2s_{T-t_k}(Z_{t_k})] + \sqrt{2h_{k+1}} B_k, \quad (4)$$

where $\{B_k\}$ are i.i.d sampled from $\mathcal{N}(O_d, I_d)$. These approximations form a simulation of the backward SDE, so the distribution of $Z_t$ is close to the distribution of $Y_t$. Denote $Q_t$ as the distribution of $Z_t$ thus $Q_t \approx P_{T-t}$ and $Q_T \approx P_0$. However, due to the existence of approximation error, there is a difference between $Q_t$ and $P_{T-t}$.

**Training** In practice, to both model time $t$ and input $\sqrt{\alpha_t} x + \sigma_t z$, a common way is training to embed time and input the embedding and noised data into the network. However, in image-generating tasks, this time embedding is always only available at finite points $\{\hat{t}_m\}_{m=1}^M$. We note that these embedding available points include discretization points $\{t_n\}_{n=1}^N$ but they are not the same, because not every embedding available points participate in the generating process (eg. DDIMs).

## 4 CONVERGENCE BOUNDS

In this section, we discuss a series of convergence bounds and summarize them into a parameterized form $EB$ from a theoretical view. Different from papers that focus on how many discretization points ensure convergence, we focus more on how the structure of stepping schedules affects the generating performance. Thus for the following convergence bounds, we ignore the detailed constant terms and terms with a higher order of time stepping size. To begin with, we give some basic assumptions on score matching loss and constraints on target distributions, which are also widely used in existing works Chen et al. [2023, 2022].

**Assumption 4.1** (Bounded score matching loss). *For the score estimator $s_t$, there exist a function $\epsilon(t)$ that satisfies for all $t \in (\delta, T]$, $\mathcal{D}_{SM}(s_t, p_t) \leq \epsilon(T - t) < \infty$.*

This assumption ensures that given noised input data and time, the network functions as an effective score estimator,

maintaining a reasonable $L^2$ score matching loss. Unlike assumptions in [Lee et al., 2022] and [Chen et al., 2022] where the bound of DSM loss is bounded uniformly, or assumptions in [Lee et al., 2023] where the relationship between bound of DSM loss and the $t$ is constrained, or unlike assumptions in [Chen et al., 2023] where the bound of DSM loss needs to fit a time-stepping schedule, we consider the relationship between bound of DSM loss and time and treat it an unchangable value then reversely fit it by selecting discretization points.

For the target data distribution $P_0$, we make the following assumptions to ensure that the forward SDE converges to Gaussian distribution Vempala and Wibisono [2019].

**Assumption 4.2** (Bounded second moment). *We assume* $\mathbb{E}_{x \sim P_0}[\|x\|^2] < \infty.$

This assumption is required for most existing convergence works of diffusion models Lee et al. [2022], Chen et al. [2023], Lee et al. [2023] and holds for almost all distributions that are worth sampling.

To convert KL divergence or TV distance between $P_\delta$ and $Q_{T-\delta}$, a widely used approach is Girsanov's theorem, which is brought to diffusion models in Chen et al. [2022]. With a constrained starting discretization point, the part that the time-stepping schedule affects is:

$$\sum_{k=0}^{N-1} {}_{all} \mathop{\mathbb{E}}_{Y_t \sim Q_t} \int_{t_k}^{t_{k+1}} \|s_{T-t_k}(Y_{t_k}) - \nabla \ln p_{T-t}(Y_t)\|^2 dt.$$

Common methods Chen et al. [2023, 2022] expand it through approximation line: $s_{T-t_k}(Y_{t_k}) \to \nabla \ln p_{T-t_k}(Y_{t_k}) \to \nabla \ln p_{T-t_k}(Y_t) \to \nabla \ln p_{T-t}(Y_t)$ and deal with every term to obtain final upper bounds. It is worth mentioning upon addressing the first approximation step via Assumption 4.1, the remaining terms demonstrate a dependence on the step size with an $O(h^2)$ order. These terms are associated with smoothness properties of the target distribution and are not related to the score estimator. Due to the same framework in the proof of bounds, their results have similar forms, which can be expressed as $EB$ in this paper. Specifically, $EB$ is defined as follows:

$$EB = C_S \sum_{k=1}^{N} h_k \epsilon(t_k) + C_L \sum_{k=1}^{N} h_k^2 L(t_k). \quad (5)$$

Here, $h_k$ denotes the k-th step size, $\epsilon$ represents the function associated with the bounds of score matching, $L(t_k)$ is a coefficient that describes the $h^2$ dependency in the convergence bounds, and $C_S, C_L$ are constants that are independent of the time-stepping schedule.

Bringing $EB$ into adaptive time-stepping schedule remains some unknown terms $C_S, C_L, \epsilon, L$. Now, we fix them one by one. $C_S$ and $C_L$ are constants for $t$ and time-stepping

schedule, thus can be treated as a hyperparameter when $EB$ is used for adjusting discretization points. We note that they merge to one parameter by considering $E_B/C_S$ and treat $C = C_L/C_S$ as a ratio.

To obtain the exact value of $\epsilon(t)$, one approach is the equivalence DAE loss by subtracting a network-constant term. As a bound of score matching loss, DAE is tight for high $t$ but slack for small $t$. Besides, there exist theoretical guarantees, for example, $\epsilon(t) = C\sigma_t^{-2} = C(1 - e^{-2t})^{-1}$ is provided by Oko et al. [2023], Gupta et al. [2023] for the case when $P$ is bounded with smooth p.d.f and polynomial samples are provided for training.

When it comes to $L$, a common approach is putting assumptions on the smoothness property. One line research Chen et al. [2022], Lee et al. [2023] makes an assumption on the Lipschitz property of the score function. This assumption also holds for many situations, such as when the target distribution is Gaussian. In this Lipschitz setting, following the convergence bound provided by Chen et al. [2022], and concentrating on components that are contingent on the time-stepping schedule, we have the following lemma.

**Lemma 4.3** (Lipschitz setting). *Suppose Assumption 4.1 and 4.2 hold, and there exist a Lipschitz score $L$ that satisfies $\nabla \ln p_t$ is $L$-Lipschitz for all $t$. Then running simulation 4 to end ($\delta = 0$) with discretization points $\{t_k\}$, distribution $Q_T = law(Z_T)$ satisfies*

$$TV(P_0, Q_T)^2$$
$$\leq C_S \sum_{k=0}^{N-1} h_{k+1} \epsilon(t_k) + C_L \sum_{k=0}^{N-1} L^2 h_k^2 + o(h^2) + C,$$

*where $o(h^2)$ owns higher order of $h_k$ and $C, C_S, C_L$ are not related to time-stepping.*

This lemma tells that $L(t)$ can be chosen through the Lipschitz constant. Due to a lack of knowledge of the target distribution, this constant is always unknown. It can also be treated as a hyperparameter in the Lipschitz setting and can also be merged into $C_L$ when $EB$ is used for adjusting discretization points.

Another line of research Chen et al. [2023] drops the Lipschitz assumption. Instead, they let backward simulation not continue to $\delta = 0$ because $\delta > 0$ also constrains the smoothness of the score function at $t = \delta$ in another way and then constrains smoothness for all $t$. In this early stopping setting, sampling error has the following bound, which tells that $L(t)$ can be chosen to $\sigma_{T-t_k}^{-4}$.

**Lemma 4.4** (Early stopping setting, Theorem 2 in Chen et al. [2023]). *Suppose Assumption 4.1 and 4.2 hold, running simulation 4 to $\delta > 0$ with time discretization points $\{t_k\}$,*

*distribution $Q_{T-\delta} = law(Z_{T-\delta})$ satisfies*

$$KL(P_\delta \| Q_{T-\delta})$$
$$\leq C_S \sum_{k=0}^{N-1} h_{k+1}\epsilon(t_k) + C_L \sum_{k=0}^{N-1} \frac{h_{k+1}^2}{\sigma_{T-t_k}^4} + o(h^2) + C,$$

*where $o(h^2)$ owns higher order of $h_k$ and $C, C_S, C_L$ is not related to time-stepping.*

In the following passage, we treat $EB$ not only as a series of convergence bounds but also as the loss function to each discretization point. Except for the reason of its theory supporting, as a loss function, $EB$ considers the balance between different performances of score estimators and stepping size. In details, term $C_S \sum_{k=1}^{N} h_k\epsilon(t_k)$ give punishment for selecting weak score estimators while term $C_L \sum_{k=1}^{N} h_k^2 L(t_k)$ avoids stepping size to be too high. For convenience, denote $E(t, s)$ as $C_S\epsilon(t)(s-t) + C_L L(t)(t-s)^2$, then $EB$ becomes $EB = \sum_{k=1}^{N} E(t_{k-1}, t_k)$ and the only term that related to $t_k$ is $E(t_{k-1}, t_k) + E(t_k, t_{k+1})$. Though there is a gap between minimizing convergence bounds and minimizing sampling error, we empirically show their consistency in Figure 2, 3 and Section 6.

## 5  TIME-STEPPING SCHEDULES

In this section, we discuss how the structure of time-stepping schedules affects convergence bounds and generating results. We note that to discuss the structure of the time-stepping schedule, the number of time discretization points and ending time $T$ need to be constrained.

Firstly in subsection 5.1, we provide evidence that normal time-stepping schedules fail to converge well both theoretically and empirically even in simple tasks. Next in subsection 5.2, we provide our adaptive time-stepping schedules and show their advances in the above situations.

### 5.1  WEAKNESS OF FIXED STEPPING SCHEDULES

Fixed time-stepping schedules mean time discretization points $\{t_k\}$ do not change even with knowledge of score matching loss. Existing stepping schedules include linear schedule ($\beta_{t_k}$ is linear to $k$ and used in Ho et al. [2020]), cosine schedule ($\overline{\alpha}_t = f(t)/f(0)$ and used in Nichol and Dhariwal [2021]) and uniform discretization points ($t_k$ is linear to $k$ and used in theoretical analysis). Due to the random variance in the training process, the score estimator at every discretization point has a variety of performances, indicating treating them with the same importance may cause weak generating performance.

We consider a simple case: generating samples from Gaussian target distribution $P = \mathcal{N}(\mu, \sigma^2)$ with $\mu = 10, \sigma^2 =$

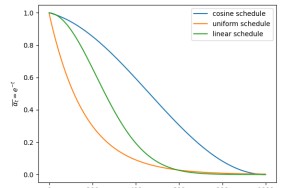 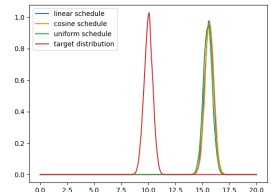

Figure 1: Weakness of fixed stepping schedules. Left: $\overline{\alpha}_t$ throughout diffusion in 3 fixed stepping schedules. Right: p.d.f of generated results for 3 fixed stepping schedules with 1000 discretization points.

0.16 and score estimator with shaking score matching loss $s_t(x) = -\frac{x - e^{-t}\mu}{\sigma^2} + 10\cos(10\pi t)$. In this situation, we run the three fixed time-stepping schedules with 1000 steps to the generated results and get results shown in Figure 1. For generating tasks, it is likely to consider how many generated samples show a high probability of target distribution. In this view, all three fixed stepping schedules fail to generate ideal samples.

However, with the given score estimators, it is not impossible to generate good samples. If discretization points are selected at $t_n = (2n - 1)/10, (n > 0, N = 5)$ (only 10 points), the generated samples follow $\mathcal{N}(9.56, 0.32)$, which is much near to target distribution and good samples can be generated, thus only 10 discretization points can generate well. The reason why 1000 points in the given time-stepping schedule can not generate well is many weak score estimators participate in the generating process in fixed stepping schedules, which causes $Q_t$ to deviate from $P_{T-t}$. This weakness can not be avoided unless discretization points are selected to adapt score matching losses.

Though this constructed situation of extremely shaking score error is specific and hardly exists in real-world situations, small shaking exists due to the independence between different time embedding. If all discretization points are fixed, it is likely to let weak score estimators participate in the generating process which decreases generating performance, especially for under-trained diffusion models.

### 5.2  ADAPTIVE TIME-STEPPING SCHEDULES

To solve the weakness of fixed time-stepping schedules, we propose to adjust them with the knowledge of score matching loss and minimize convergence bounds mentioned in section 4. In the practice of diffusion models, there are two settings for embedding time: continuous-time diffusion (all times are embedding available points) and discrete-time diffusion (finite embedding available points). For each setting, we give methods to achieve adaptive time-stepping schedules.

**Continuous-time diffusion** In the realm of continuous-time diffusion, we delve into a framework where time is embedded at every point within the interval $[0, T]$. This characteristic makes it possible for us to use a score estimator at each moment in time for generating. Within this setting, our purpose is to find the exact locations for each stepping point, thereby enhancing the generative performance of the diffusion models. In this work, we improve the generating performance by minimizing the above convergence bounds. With knowledge of score matching bounds and smoothness variance, the only unknown terms that matter in $EB$ are $C_S$ and $C_L$, which can turn to one hyperparameter $C = C_S/C_L$ by equivalently minimizing $EB/C_S$. Fortunately, due to the form of $EB$, the only $t_k$ related term is $E(t_{k-1}, t_k) + E(t_k, t_{k+1})$ which makes its gradient can be calculated easily. Following the above observation, we propose a gradient-based adjusting stepping schedule (GA, Algorithm 1) to run gradient descent for discretization points with loss function $EB$.

---

**Algorithm 1** Gradient-based Adjusting stepping schedule

**Input:** initial discretization points $\{t_k\}$, score matching bound $\epsilon(t)$, smoothness variance $L(t)$, ratio $C$, learning rate $\eta$.
**repeat**
  **for** $k$ **in** $[N-1]$ **do**
    Calculate $g_k = -\epsilon(t_{k+1}) + 2(t_k - t_{k+1})L(t_{k+1}) + \frac{1}{2\Delta t}\left[E(t_{k-1}, t_k + \Delta t) - E(t_{k-1}, t_k - \Delta t)\right]$.
  **end for**
  **for** $k$ **in** $[N-1]$ **do**
    Update $t_k = t_k - \eta g_k$.
  **end for**
**until** convergence

---

In the implementation of GA, the initial plan is flexible and can be chosen as existing fixed stepping schedules. The selection of the score matching bound $\epsilon(t)$ is equally versatile and can be grounded in either theoretical guarantees or specific measurement methods tailored to the task at hand, such as when the target distribution is known. Additionally, the DAE error, suitably normalized by a constant, stands out as a viable choice, serving as an upper bound for the score matching loss. The determination of the smoothness variance $L(t)$ depends on theoretical settings. If the prior information sheds light on the Lipschitz properties of score functions, for instance, when the target distribution follows a Gaussian pattern, the Lipschitz setting becomes a pertinent choice. Also, the setting of early stopping is global and the corresponding smoothness variance is known. The ratio $C$, which refers to $C_L/C_S$, acts as a positive hyperparameter in Algorithm 1. This is due to the unknown inner structure of convergence bounds. We note that the real value of this ratio in theoretical bounds even exhibits variability with slight changes in algebraic considerations.

GA introduces an approach for selecting better discretiza-

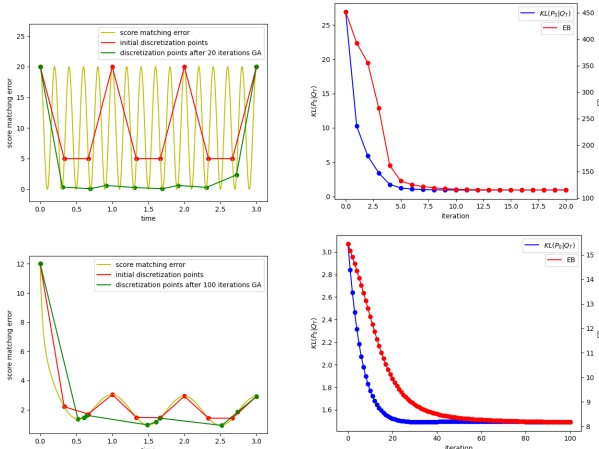

Figure 2: Two examples for running GA (Algorithm 1) for 20/100 iterations on the Gaussian target, with heavily/slightly shaking score error and 10 discretization points in $[0, 3]$. Left figures: selected discretization points after running GA for iterations and origin normal selected points. Right figures: $EB$ and $KL$ decreasing with increasing iterations (0 iterations for fixed schedule).

tion points for continuous-time diffusion models. Due to the limited $t_0$ and $t_N$, term $C_S \sum_{k=1}^{N} h_k \epsilon(t_k)$ in $EB$ makes GA tends to select points with lower score matching bounds. For the shaking score error mentioned in 5.1, the adjusted stepping schedule adjusted by GA and its generating performance is shown on the top two graphs of Figure 2, where the adjusted discretization points are all points with small score error (except constrained points). In this situation, $KL(P_0, Q_T)$ decreases from 26.87 to 1.13 only within 20 iterations. However, GA does not always select points with low score matching loss, it also considers the size of stepping in term $C_L \sum_{k=1}^{N} h_k^2 L(t_k)$. This is shown in the below two graphs of Figure 2, where some points with not that low score error are selected to maintain the stepping size to stay low.

**Discrete-time diffusion** In the realm of discrete-time diffusion, time is only embedded at finite embedding available points $\{\hat{t}_m\}_{m=1}^{M}$. This limits us to use only finite choices of discretization points for generating and score matching losses are also discrete. Within this setting, our purpose is to choose $N$ discretization points among them for generating, thereby enhancing the generative performance of the diffusion models. Similarly, we also focus on minimizing above convergence bounds and propose to achieve it by iteration greedy optimization. Different from the continuous-time setting, we select the best position (with the lowest $EB$) among candidate points discretely in each iteration, rather than continuously adjusting $t_k$ to find a better position. This allows us to indeed get a discretization schedule with lower $EB$ in each iteration. The core idea is that the best stepping

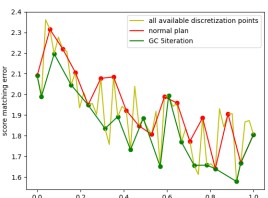 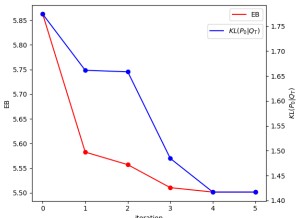

Figure 3: An example for running Algorithm 2 for 5 iterations on the constructed Gaussian target, random score error, and discretization points from uniform 51 points in $[0, 1]$ select 17. Left figure: selected discretization points after running GC for 5 iterations vs. origin normal selected points. Right figure: EB alone with KL divergence decreasing with increasing iterations

schedule cannot be adjusted. Even though the converged result may be just local optima but not global minima, it is still better than the initial schedule. This approach also satisfies that $EB$ is minimized in each iteration, thus the optimized schedule is always better than the initial schedule on $EB$. Utilizing notions $E(t, s)$ defined in section 4, GC can be stated as Algorithm 2.

---

**Algorithm 2** Greedy Choosing time-stepping points

---

**Input:** embedding available points $\{\hat{t_m}\}$, initial discretization points $\{t_k\}$, score matching bound $\epsilon(t)$ (only available at $t = \hat{t_m}$), smoothness variance $L(t)$, ratio $C$
**repeat**
    **for** $k$ in $[N-1]$ **do**
        Find all candidates $A = (t_{k-1}, t_{k+1}) \cap \{\hat{t_m}\}$
        Set $t_k = \arg\min_{t \in A} E(t_{k-1}, t) + E(t, t_{k+1})$
    **end for**
**until** convergence

---

In the implementation of GC, the initial plan is flexible and can follow the selecting methods in Song et al. [2020a] to be leading. The choice of smoothness variance depends on the Lipschitz setting or early stopping setting, while ratio $C$ acts as a positive hyperparameter, which is the same as implementation in GA. Different from GA, in the discrete-time setting, the target distribution is always unknown, indicating score matching bound needs to be calculated by DAE loss. Fortunately, DAE loss is always an upper bound for score matching loss and is tight when $t$ is high. For small $t$, GC tends to not choose discretization points because DAE loss is high, but this is caused by a high gap between DAE loss and score error. To solve this issue, we limit the candidates to not changing a lot from the initial plan in practice.

GC improves generating performance for well-trained diffusion models by intelligently selecting discretization points to expedite the generation process. Unlike fixed schedules that might overlook the intricate dynamics between model decisions and the target distribution, GC fits them by min-

imizing convergence bounds. It strategically picks points that do not decrease much output quality while accelerating generating, adeptly avoiding the common pitfalls of rigid scheduling. This approach ensures that GC-driven generating processes are not just fast but also finely attuned to the desired outcomes, delivering swift and high-quality results in a streamlined and efficient manner.

GC also works for under-trained diffusion models, where score matching losses vary at different times. This happens because the time embeddings are independent of each other. When choosing discretization points with a fixed schedule, both the weaker and the stronger score estimators get involved in creating the output. It's like having a team with both new and experienced players contributing equally, which results in average performance overall. Acting as a coach, GC picks the discretization points carefully so that stronger score estimators participate in generating, while weaker estimators do not. By doing so, GC ensures that the overall performance of the diffusion model is enhanced (shown in Figure 3), much like a team would perform better if guided by a strategic coach who knows when to play each member. This approach makes diffusion models able to generate high quality at an earlier stage of training.

# 6 EXPERIMENTS

In this section, we provide more empirical evidence that GC improves the generating quality for fixed stepping schedules with the same number of steps and the same starting time. For well-trained models, we first test GC on the CI-FAR10 dataset for the well-trained model provided on hugging face google/ddpm-cifar10-32 and provide FID scores. Then, we test GC on the CELEBA dataset for the model provided on google/ddpm-celebahq-256 and provide generated examples in 10 generating steps, comparing with the DDIM fixed schedule. For under-trained models, to make advances more intuitive, we train a DDIM model at the MNIST dataset and show generated results in 5 generating steps in the three time-stepping schedule. On the hardware front, both datasets were trained on one NVIDIA A30 GPU. Model performance was evaluated based on the FID score of 50,000 generated images against real-world images Heusel et al. [2017], Jolicoeur-Martineau et al. [2020]. The code is available at https://github.com/cyzkrau/AdaptiveSchedules.

We first test the performance boost GC provides to a well-trained model. To avoid the influence caused by a gap between DAE and DSM, we set constraints for selected discretization points from changing too much in each iteration. Following this, in Table 1, we evaluate the FID scores of samples generated by the same model provided in hugging face, with the stepping schedule provided in DDIM and the GC-adjusted one. We test GC for $0-5$ iterations (0 iterations for DDIM fixed schedule) in both the Lipschitz score setting ($CL = 1$) and the early stopping setting ($CL(T-t) = \sigma_t^{-4}$).

Table 1: For given DDIM model google/ddpm-cifar10-32, FID scores for stepping schedule after 1-5 iteration of Algorithm 2 against stepping schedule in DDIM on CIFAR10(32x32). In Algorithm 2, we set the bound of score matching loss $\epsilon$ to be calculated by DAE loss. $L$ is tested under Lipschitz setting bounds (constant) and early stopping setting bounds (the reciprocal of the noise variance).

| Stepping schedule \ # discretization points | 5 | 10 | 20 | 50 | 100 |
|---|---|---|---|---|---|
| DDIMs (Cosine schedule) | 110.49 | 47.06 | 22.52 | 11.91 | 9.06 |
| GC - Lipschitz - 1 iteration | 97.63 | 44.10 | 21.68 | 11.76 | 9.06 |
| GC - Lipschitz - 2 iterations | **91.13** | 42.85 | 21.44 | 11.81 | **9.02** |
| GC - Lipschitz - 3 iterations | 91.54 | 42.39 | **21.30** | **11.70** | 9.16 |
| GC - Lipschitz - 4 iterations | 93.88 | **42.24** | 21.47 | 11.91 | 9.16 |
| GC - Lipschitz - 5 iterations | 94.54 | 42.73 | 21.56 | 11.95 | 9.14 |
| DDIMs (Cosine schedule) | 110.49 | 47.06 | 22.52 | 11.91 | 9.06 |
| GC - stopping - 1 iteration | 98.37 | 43.98 | 21.62 | **11.69** | 9.06 |
| GC - stopping - 2 iterations | 90.90 | 42.54 | 21.55 | 11.73 | **8.90** |
| GC - stopping - 3 iterations | 87.59 | **41.43** | **21.23** | 11.85 | 9.08 |
| GC - stopping - 4 iterations | 87.06 | 42.13 | 21.52 | 12.07 | 9.14 |
| GC - stopping - 5 iterations | **86.93** | 42.76 | 21.58 | 11.94 | 9.23 |

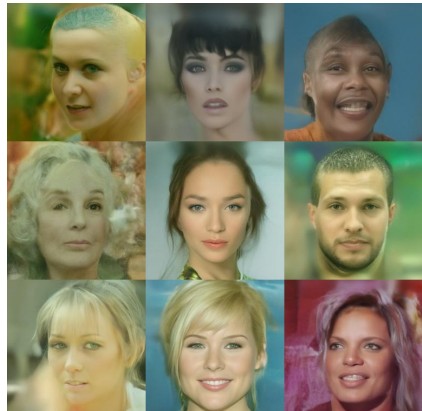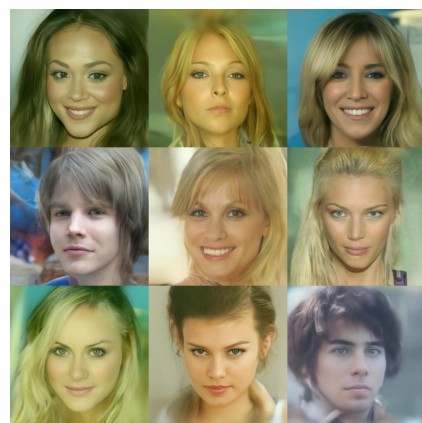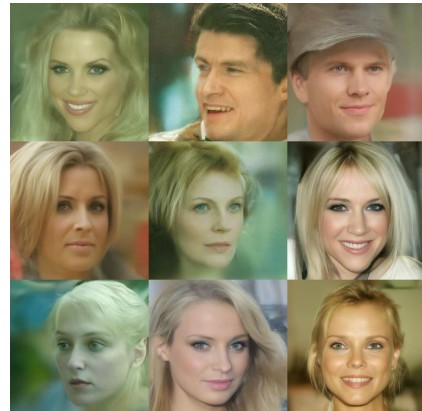

Figure 4: For given DDIM model google/ddpm-celebahq-256, 9 generated images under 10 steps for stepping schedule in DDIM fixed schedule (left) and 5 iterations GC schedule in Lipschitz setting (middle) and early stopping setting (right). In Algorithm 2, we set the bound of score matching loss $\epsilon$ to be calculated by DAE loss. $L$ is tested under Lipschitz setting bounds (constant) and early stopping setting bounds.

We observe that the GC produces higher quality samples than the original stepping schedule with the same sampling steps and the final $t_N$. We also observe that GC shows much improvement for smaller steps (especially 5 steps) in the generating process, this may be because every discretization point becomes more important with decreasing steps. This demonstrates that the well-trained diffusion model enhanced by our adaptive stepping schedule contributes to the improvement of sample quality, especially in cases when a few steps are used.

We also test GC in the CELEBA dataset to show how it enhances the image generation performance. As shown in Figure 4, the images generated with a fixed schedule exhibit more noticeable flaws in facial edge details compared to those generated with GC-adjusted schedules. Following empirical observations by Wang et al. [2024], who suggest

that for the backward process of diffusion models, initial steps are crucial in establishing the overall structure while later steps work on refining details, this difference could be attributed to GC's ability to balance the selection of small and large time steps in a few choices, thereby handling the trade-off between local details and overall structure under constraints, resulting in higher-quality outputs.

We then give generated samples to show GC's improvement for an under-trained model. In Figure 5, we show the quality of samples generated in 5 steps by the same under-trained model which is trained for 100 epochs on the MNIST dataset, along with the time-stepping schedule provided in DDIM and GC adjusted discretization points. We show the generated results from GC converged discretization points and compare them with the DDIM fixed schedule. As expected, enhanced by GC, the model generates more reason-

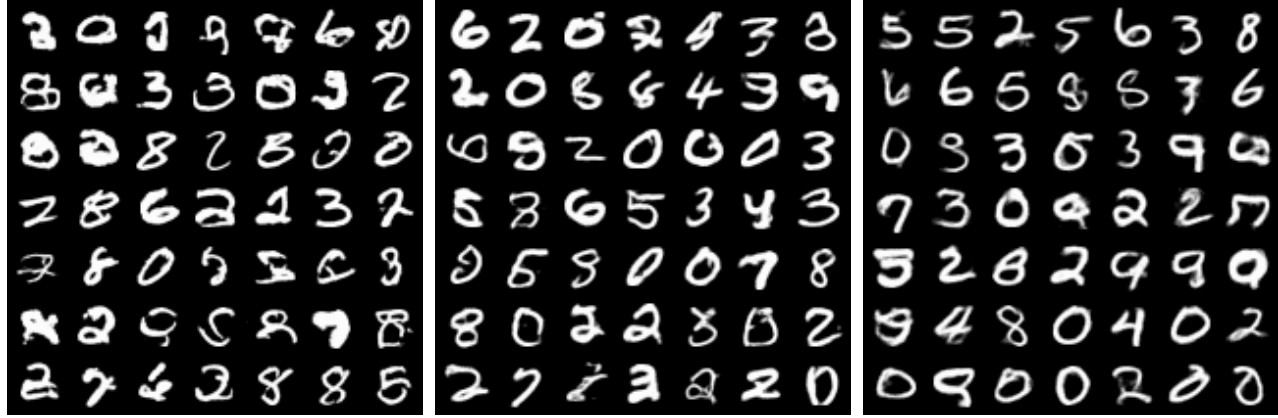

Figure 5: For DDIM model trained for 100 epochs on MNIST, 49 generated samples for stepping schedule in DDIM fixed schedule (left) and 5 iterations GC schedule in Lipschitz setting (middle) and early stopping setting (right). In Algorithm 2, we set the bound of score matching loss $\epsilon$ to be calculated by DAE loss. $L$ is tested under Lipschitz setting bounds (constant) and early stopping setting bounds.

able and denoised samples, which proves the improvement GC brings to under-trained models.

# 7   CONCLUSION

We considered the problem of selecting discretization points to adapt the score matching loss and the smoothness property on the generating process of diffusion models. For continuous-time diffusion, we propose GA for adjusting and GC for choosing in discrete-time diffusion. GA avoids the weakness of fixed schedules and can generate shaking score errors. GC improves generating performance for both well-trained and under-trained models, especially with a few discretization points.

Our methods can be further improved in several directions: (1) tighter convergence bounds: our methods aim to minimize sampling error, but existing upper bounds include gaps to the target, and their instruction on ratio $C$ are highly different; and (2) adaptive training schedule: we only consider the generating process and treat the model as an unchangeable part, but the training process can also be adaptive.

### Acknowledgements

This work was supported in part by NSFC No. 62222117.

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

# A ADDITIONAL EXPERIMENT IMPLEMENTATION DETAILS

In this section, we include more details about experiments.

For all Gaussian target tasks, the target distribution are on $\mathcal{N}(10, 0.16)$ on $\mathbb{R}$. Score error is selected to be constant with

1. Figure 1 and Figure 2 top: $10 + 10\cos(10\pi t)$;
2. Figure 2 below: $\frac{1}{1.1 + e^{-2t}} + \cos(2\pi t) + 1$;
3. Figure 3: $\frac{1}{1+t} + 1$ plus $[0, 1]$ random rv.

Score matching bounds $\epsilon$ (also shown as smb in codes) are settled to be the square of score error. Smoothness variance is select in the Lipschitz setting, where the Lipschitz constant at $t$ is $\sigma_t^{-2}$. Ratio $C$ is 1 for all GA tests and 10 for the GC test in Figure 3. For evaluation, we use both optimized $EB$ (of course decreasing) and $KL(P_0\|Q_T)$ (as sampling error), and they show consistency in all Gaussian target tasks. We also note that $KL$ divergence is calculated by generated mean and variance, rather than just sampling. In our setting of constant score error, all steps are linear because

$$Z_{t_{k+1}} = Z_{t_k} + h_{k+1}\left[Z_{t_k} + 2s_{T-t_k}(Z_{t_k})\right] + \sqrt{2h_{k+1}}B_k \tag{6}$$

$$= Z_{t_k} + h_{k+1}\left[Z_{t_k} - \frac{2Z_{t_k} - 2e^{t_k - T}\mu}{1 - e^{2t_k - 2T}} - 2\sqrt{\epsilon(t_k)}\right] + \sqrt{2h_{k+1}}B_k. \tag{7}$$

Thus every $Q_{t_k}$ is also Gaussian. Denote $Q_{t_k}$ as $\mathcal{N}(\mu_k, \sigma_k^2)$ then $\mu_k$ and $\sigma_k^2$ satisfy:

$$\mu_{k+1} = \left[1 - \frac{1 + e^{2t_k - 2T}}{1 - e^{2t_k - 2T}}h_{k+1}\right]\mu_k + \frac{2e^{t_k - T}}{1 - e^{2t_k - 2T}}h_{k+1}\mu - 2h_{k+1}\sqrt{\epsilon(t_k)},$$

$$\sigma_{k+1}^2 = \left[1 - \frac{1 + e^{2t_k - 2T}}{1 - e^{2t_k - 2T}}h_{k+1}\right]^2\sigma_k^2 + 2h_{k+1}.$$

We use these equations to calculate the theoretical distribution of $Q_T$ then derive $KL(P_0\|Q_T)$.

For image-generating tasks, there is a difference in notions. At $k$-th step in forward process, it is $t_{N-k}$ and score estimator $s_{N-k}$ in this paper, but noise variance $\overline{\alpha_k}$ and u-net $\epsilon_\theta(\cdot, k)$ in codes and some other papers Ho et al. [2020], Song et al. [2020a], Nichol and Dhariwal [2021]. Their connection is $t_{N-k} = -\ln\sqrt{\overline{\alpha_k}}$. For implementation of experiments in image-generating tasks, score matching bounds are derived through DAE loss. We note that there is a difference between DAE loss and the loss function for training diffusion models. In the notion of $\alpha$, DAE loss should be calculated through

$$\mathcal{D}_{AE}(s_k, P, k) = (1 - \overline{\alpha_k})^{-1}\mathbb{E}_{x \sim P, \epsilon \sim \mathcal{N}(O,I)}\|\epsilon_\theta(\sqrt{\overline{\alpha_t}}x + \sqrt{1 - \overline{\alpha_t}}\epsilon) - \epsilon\|^2.$$

For smoothness variance $L$, we test in both Lipschitz score setting and early stopping setting. For the Lipschitz score setting, we assume an inner Lipschitz constant $\tilde{L}$ for all score functions and set $C$ to let $C\tilde{L} = 1$, which is equivalent to setting $C = 1, L = 1$ when running GC.

# B PROOF FOR LEMMA 4.3

In this section, we provide the proof of Lemma 4.3. This lemma is derived through the framework of Chen et al. [2022], where bound on the discretization error, i.e.

$$\sum_{k=0}^{N-1} \mathop{\mathbb{E}}_{all \ Y_t \sim Q_t} \int_{t_k}^{t_{k+1}} \|s_{T-t_k}(Y_{t_k}) - \nabla\ln p_{T-t}(Y_t)\|^2 dt,$$

is the only part that relies on the time-stepping schedule. Considering each term without integration and expectation, we have

$$\|s_{T-t_k}(Y_{t_k}) - \nabla\ln p_{T-t}(Y_t)\|^2 \le 3\|s_{T-t_k}(Y_{t_k}) - \nabla\ln p_{T-t_k}(Y_{t_k})\|^2$$

$$+ 3\|\nabla\ln p_{T-t}(Y_{t_k}) - \nabla\ln p_{T-t_k}(Y_{t_k})\|^2 + 3\|\nabla\ln p_{T-t}(Y_{t_k}) - \nabla\ln p_{T-t}(Y_t)\|^2$$

$$\le 3\epsilon(t_k) + 3\|\nabla\ln\frac{p_{T-t}}{p_{T-t_k}}(Y_{t_k})\|^2 + 3L^2\|Y_t - Y_{t_k}\|^2.$$

Taking expection and considering $|t - t_k| < h_{k+1}$ for all $t \in [t_k, t_{k+1}]$, following proofs for DDPM in Chen et al. [2022], we obtain

$$\mathop{\mathbb{E}}_{all \ Y_t \sim Q_t} \|s_{T-t_k}(Y_{t_k}) - \nabla \ln p_{T-t}(Y_t)\|^2 \leq C_S \epsilon(t_k) + \frac{C_L}{2} L^2 h_{k+1} + O(h_{k+1}^2).$$

Taking integration, the first term becomes $C_S \epsilon(t_k) h_{k+1}$, the second term becomes $C_L L^2 h_{k+1}^2$ and the third term becomes $o(h_{k+1}^2)$. Summing them up and considering the difference between $P_T$ and $Q_0$ term into $C$, we get Lemma 4.3.