# OpenReview forum: "Adaptive Time-Stepping Schedules for Diffusion Models"
_auai.org/UAI/2024/Conference — UAI 2024 poster_

### Official Review · Reviewer_p641 · 2024-02-28

**Q2-1 Originality-Novelty:** 2
**Q2-2 Correctness-Technical Quality:** 2
**Q2-5 Clarity Of Writing:** 1

**Q1 Summary And Contributions:**

The authors propose adaptive time-stepping schedules for diffusion models. Using known error bounds, these are optimized for using gradient descent for continuous diffusion, and for discrete diffusion a greedy algorithm is used. These methods are tested in experiments showing improved generation ability.

**Q2-3 Extent To Which Claims Are Supported By Evidence:**

1: Poor: the authors fail to convincingly backup their main claims (e.g., if the experimental evaluation is flawed, proofs are lacking or invalid, references are missing, assumptions are not realistic, not specified, or not motivated).

**Q2-4 Reproducibility:**

2: Fair: key resources (e.g. proofs, code, data) are unavailable but key details (e.g. proof sketches, experimental setup) are sufficiently well-described for an expert to confidently reproduce the main results.

**Q3 Main Strengths:**

* Diffusion models are highly popular and improved generation abilities by optimizing the temporal discretization is an interesting line of research.

**Q4 Main Weakness:**

* The abstract claims that the methods are tested extensively, but the authors only test on MNIST and CIFAR10, which are not relevant benchmarks for modern generative models anymore.
* The figures are unreadable.
* There are notational issues. E.g., I don't understand what's going on in the second equation of section 3. Is $a$ a function there $G$ is not defined in equation 2. "$law$" is (AFAIK) undefined.
* Further unclarities. On p.4 it's claimed that Assumption 4.1. exhibits an $O(h^2)$ dependence on the step size. This is not clear.
* The introduction of the methods and experiments is unclear. Some details are left out.

**Q5 Detailed Comments To The Authors:**

* Is there a proof for Theorem 4.3.?
* How is $\epsilon$ calculated in the experiments?

**Q9 Complying With Reviewing Instructions:**

Yes

---

> ### Author Rebuttal · Authors · 2024-04-05
>
> Thank you for your thorough review and constructive comments. All your concerns have been carefully addressed as below. The manuscript is carefully revised accordingly. We sincerely hope our responses fully address your questions.
>
> **Q1:** Experiments of more benchmarks.
>
> **A:** Thanks. We respectfully note that besides CIFAR10 and MNIST, we also provided experiments on CELEBA in the supplementary material.
>
> Following your suggestions, we conducted additional experiments on CELEBA during the rebuttal session. We tested the fixed cosine schedule and its GC-enhanced schedule (in both Lipschitz and early stopping settings) on CELEBA. The selected diffusion model is from huggingface google/ddpm-celebahq-256 (well-trained). We used FID1k to measure the model performance. The experimental results are in agreement with our method and similar to FID results on CIFAR10:
>
> | #Steps | 5 | 10 | 50 | 100 |
> | :---: |:---:| :---:|:---:| :---:|
> | Cosine schedule | 106.67 | 74.68 | 61.13 | 59.20 |
> | GC Lipschitz 5 iterations | 102.55 | 73.70 | 60.19 | 60.08 |
> | GC early stop 5 iterations | 102.96 | 74.20 | 60.10 | 58.70 |
>
> **Q2:** Explanation of notions.
>
> **A:** Thanks. We clarify as follows:
> - **The second equation in Section 3:** this is the analytical form of the forward process (the first equation in Section 3). In the following formula, $f_{(a)}$ denotes a scaled density function derived from $f$, where $a$ is used to indicate that form $f_{(a)}$ is a scaling of the density function $f$.
> - $G$ in Equation 2: we use $G$ to denote the Gaussian distribution $\mathcal(O_d, I_d)$. This is because we use $P$ (the upper letter) to denote an distribution and $p$ (corresponding lower letter) to denote density function. We used $g$ denote its density function of $\mathcal(O_d, I_d)$ in the second equation of section 3, thus $G$ for $\mathcal(O_d, I_d)$. For easier to understand, we will change it to $\mathcal(O_d, I_d)$ in the revised paper.
> - The meaning of“law”: we use $Q_T=law(Z_T)$ to denote the distribution of $Z_T$ as $Q_T$. This notion is also used in [1, 2].
>
> **Q3:** Proof for Theorem 4.3 and why $O(h^2)$ dependence on the step size without score approximation term.
>
> **A:** Thanks. We clarify this by the following, detailed proof.
>
> By utilizing Theorem 8 in [1] and the approximation argument part of Theorem 9 in [1], $TV(P_0, Q_T)^2$ can be upper bounded by
> $$\sum_{k=0}^{N-1}\mathbb{E}\int_{t_{k}}^{t_{k+1}}C_1\|s_{T-t_{k}}(Y_{t_{k}})-\nabla\ln p_{T-t}(Y_t)\|^2dt +C_2. $$
> Through approximation line $s_{T-t_k}(Y_{t_k})\to\nabla\ln p_{T-t_k}(Y_{t_k})\to  \nabla\ln p_{T-t_k}(Y_t)\to \nabla\ln p_{T-t}(Y_t)$, we have
> $$
> TV(P_0, Q_T)^2\leq \sum_{k=0}^{N-1}\mathbb{E}\int_{t_{k}}^{t_{k+1}}3C_1\|s_{T-t_{k}}(Y_{t_{k}})-\nabla\ln p_{T-t_k}(Y_{t_k})\|^2
> $$
> $$~~~~~~~~+3C_1\|\nabla\ln p_{T-t_k}(Y_{t_k})-\nabla\ln p_{T-t_k}(Y_t)\|^2$$
> $$~~~~~~~~+3C_1\|\nabla\ln p_{T-t_k}(Y_t)-\nabla\ln p_{T-t}(Y_t)\|^2dt +C_2. $$
> Through Assumption 1, the first term is bounded by $C_S\sum_{k=1}^Nh_k\epsilon(t_k)$.
>
> Through Lipschitz assumption, the second term (for k) is bounded by $3C_1L_{t_k}^2\mathbb{E}\|Y_{t_k}-Y_t\|^2$ which can be then bounded by $3C_1L_{t_k}^2dh+O(h^2)$ through Lemma 11 in [1]. With integration and summing up with $k$, it becomes $C_3\sum_{k=1}^Nh_k^2L_{t_k}^2+o(h_k^2)$.
>
> The third term for $k$ is bounded through Lemma C.12 in [3] to $324C_1L_{t_k}^2dh_k+O(h_k^2)$. With integration and summing up with $k$, it becomes $C_4\sum_{k=1}^Nh_k^2L_{t_k}^2+o(h_k^2)$.
>
> Summing the three parts up gets Theorem 4.3. Notice that except the first term, which is the only term related to score approximation, other terms are $O(h^2)$.
>
> **Q4:** How is $\epsilon$ calculated in the experiments?
>
> **A:** Thanks. If the score matching error is unknown, $\epsilon$ is calculated by the denoising auto-encoder (DAE) loss, as defined in Equation (2) and in the second paragraph of Section 5.2. In the practice, it is calculated by
>
> $$\sum_{n=1}^{N}(1-\overline{\alpha_k})^{-1}\|\epsilon_\theta(\sqrt{\overline{\alpha_t}}x_n+\sqrt{1-\overline{\alpha_t}}\epsilon_n)-\epsilon_n\|^2,$$
>
> where $x_n$s are from $P$ and $\epsilon_n$s are from $\mathcal{N}(O,I)$, $N$ is the number for simulating expection and $\alpha_t=e^{-t}$. It can be calculated for all $t$, thus score matching bounds known for all $t$ for continuous-time diffusion.
>
> **Q5:** Minor issues in presentation: figures, introduction of the methods and experiments.
>
> **A:** Thanks and addressed.
>
> Reference:
>
> [1] Sitan Chen, Sinho Chewi, Jerry Li, et al. Sampling is as easy as learning the score: theory for diffusion models with minimal data assumptions. ICLR. 2023.
>
> [2] Joe Benton, Valentin De Bortoli, Arnaud Doucet, et al. Linear convergence bounds for diffusion models via stochastic localization. ICLR. 2024.
>
> [3] Holden Lee, Jianfeng Lu, and Yixin Tan. Convergence for score-based generative modeling with polynomial complexity. In NeurIPS. 2022.

---

### Official Review · Reviewer_Snxt · 2024-03-15

**Q2-1 Originality-Novelty:** 3
**Q2-2 Correctness-Technical Quality:** 3
**Q2-5 Clarity Of Writing:** 3

**Q1 Summary And Contributions:**

The paper introduces two algorithms designed for selecting discretized time points in the learning of diffusion models. The core idea is selecting time points in which the error bound on the score function is minimized. Using synthetic toy examples and real-world datasets/models, the paper demonstrates the potential drawbacks of fixed time-stepping schedules, as well as the improvements in the generation process after applying the proposed algorithms.

**Q2-3 Extent To Which Claims Are Supported By Evidence:**

3: Good: the main claims are supported by convincing evidence (in the form of adequate experimental evaluation, proofs, (pseudo-)code, references, assumptions).

**Q2-4 Reproducibility:**

2: Fair: key resources (e.g. proofs, code, data) are unavailable but key details (e.g. proof sketches, experimental setup) are sufficiently well-described for an expert to confidently reproduce the main results.

**Q3 Main Strengths:**

1.	The studied problem is of great interest and importance.
2.	The key idea of the proposed solution is intriguing and innovative (to the best of my knowledge).
3.	The demonstration using toy examples provides valuable insights. Experiment results for Algorithm 2 are particularly noteworthy.

**Q4 Main Weakness:**

1.	The paper is challenging to read, and the presentation could be improved. The central idea is effectively conveyed, but some technical details were difficult to follow or verify. See detailed comments
2.	Regarding Algorithm 2: The effectiveness of Algorithm 2 hinges on the tightness of the error bound. From what I comprehend, there's no theoretical assurance that the algorithm will consistently yield improvements. Additionally, the demonstrated enhancement diminishes as the number of time points decreases.
3.	Regarding Algorithm 1 (continuous diffusion): some technical details of Algorithm 1 seem incorrect, but I may have a misunderstanding due to insufficient explanation (see detailed comments).  Additionally, there is no demonstration of algorithm 1 with real-world data, which raises concerns about the practical applicability of this algorithm.

**Q5 Detailed Comments To The Authors:**

1.	Section 3 “Forward process”  - The presentation of the Ornstein-Uhlenbeck process is very technical and lacks motivation. Specifically, the definition of $f_{a}(x)$ is difficult to understand.   Please provide references for previous papers that link between diffusion models and the Ornstein-Uhlenbeck process. I found the reference  “Using Ornstein–Uhlenbeck Process to understand Denoising Diffusion Probabilistic Model and its Noise Schedules” by J. E., & Lin, Y. T. (2023)). This reference explains that the Ornstein-Ohlenbeck (OH) process is a time-homogeneous continuous-time Markov process with analytical solution, and shows that denoising a DDPM can be presented as an OH process observed at non-uniformly sampled discrete time.
2.	Page 4: the following sentence is unclear  “Only for continuous-time diffusion, we consider $\epsilon(t)$ to be known” Why is it reasonable to assume that $\epsilon(t)$ is known?
3.	“For the target data distribution $P_0$, we make the following assumptions to ensure that the forward SDE converges to Gaussian distribution.”  Why is Assumption 4.2 needed for the forward process to converge to the Gaussian distribution?  Please provide a reference to a paper that proves this.
4.	Please add a number to the formula before (5), which describes the part that the time-stepping schedule affects.   In this formula, to what distribution does the expectation $\mathbb{E}$ refer?
5.	Suggested fix: "for time and stepping schedule” ==> “… for time-stepping schedule”
6.	Suggested fix: “There exists theoretical guarantees” => “There exist theoretical guarantees”
7.	Formula 5 will be easier to follow if all terms will be (re)defined after it.
8.	Please provide references for Theorems 4.3 and 4.4.
9.	Page 6: “due to the form of EB, the only $t_k$ related term is $E(t_{k−1}, t_{k})$”. This sentence seems to contradict the sentence on page 5 “the only term that related to $t_k$ is $E(t_{k−1}, t_k)+E(t_k, t_{k+1})$”. The latter seems correct.
10.	If I understood correctly Algorithm 1, $g_k$ is the $\frac{\partial EB}{\partial t_k}$. Please provide the complete mathematical derivation of $g_k$ from $EB$.
11.	Page 6: “such as when the target distribution is known.”, “when the target when the target distribution follows a Gaussian pattern”  - can you provide examples of real-world cases in which the target distribution is known and yet diffusion models are helpful?
12.	Supplementary Material B: can you add the results of a diffusion model with fixed-schedule on the CELEBA benchmark?
13.	Can you provide a demonstration for algorithm 1 (Continuous diffusion) on a real-world dataset?

**Q9 Complying With Reviewing Instructions:**

Yes

---

> ### Author Rebuttal · Authors · 2024-04-05
>
> Thank you for your patience, as well as your constructive comments and kind support! All your concerns have been carefully addressed below, including those on Theorems 4.3 and 4.4. The manuscript is carefully revised accordingly. We sincerely hope our responses fully address your questions.
>
> **Q1:** The effectiveness of Algorithm 2 hinges on the tightness of the error bound.
>
> **A:** Thanks. We respectfully note that the error bound gives significant inspirations for designing algorithm, though we also fully agree that it could be not tight. Inspiring from upper/lower bounds has been a successful strategy, such as we have seen in variational Bayesian methods which minimise the evidence lower bound (ELBO). Moreover, our experiments are in full agreement that the sampling error does decrease with EB, as shown in Figure 3. Inspired by these, we propose Algorithm 2 to minimize EB.
>
> **Q2:** Real-world demonstration for Algorithm 1.
>
> **A:** Thanks. Following your suggestions, we conducted a real-world experiment during the rebuttal session.
>
> As shown in the referenced work [6], the training guarantees for the score matching error of diffusion models exhibit an order of $O(\sigma_t^{-2})$, where $\sigma_t$ represents varience of time $t$. These guarantees can serve as preliminary score matching bounds. Considering both the Lipschitz setting and the early stopping setting, and taking the linear and cosine schedules as initial schedules, the Genetic Algorithm (GA) generates four additional fixed stepping schedules. With initial two schedules, we trained diffusion models on MNIST and calculated FID1k scores. The experimental results are largely consistent with our algorithm:
>
> | #Steps | Without GA | With GA Lipschitz | With GA early stopping |
> | :---: |:---:| :---:| :---:|
> | Cosine schedule | 22.18 | 21.91 | 22.04 |
> | Linear schedule | 22.84 | 22.78 | 23.22 |
>
> **Q3:** Please provide references for previous papers that link between diffusion models and the Ornstein-Uhlenbeck process.
>
> **A:** Thanks and addressed. [7] first gives the equivalence between DDPM and the Ornstein-Uhlenbeck process. Other papers include [2, 4, 5]. We will duly cite and discuss them.
>
> **Q4:** Why is it reasonable to assume score matching bounds known for continuous-time diffusion?
>
> **A:** Thanks. Score approximation is directly available, thus one score matching bounds can be denoising auto-encoder (DAE) loss by
> $$\sum_{n=1}^{N}(1-\overline{\alpha_k})^{-1}\|\epsilon_\theta(\sqrt{\overline{\alpha_t}}x_n+\sqrt{1-\overline{\alpha_t}}\epsilon_n)-\epsilon_n\|^2,$$
> where $x_n$s are from $P$ and $\epsilon_n$s are from $\mathcal{N}(O,I)$, $N$ is the number for simulating expection and $\alpha_t=e^{-t}$. It can be calculated for all $t$, thus score matching bounds known for all $t$ for continuous-time diffusion.
>
> **Q5:** Why is Assumption 4.2 needed for the forward process to converge to the Gaussian distribution?
>
> **A:** Thanks. Assume the target distribution $P$ has an infinite second moment. Then, the distribution $P_t$, which is the distribution at time $t$ of the forward process, has an infinite second moment. This means that $E_{X_t\sim P_t, Y\sim N(O_d, I_d)}\|X-Y\|^2$ are also infinite (thus forward process cannot converge to Gaussian). This result is stated in [1, 2] and analyzed in [3]. We have duly cited them in the revised paper.
>
> **Q6:** What does the expectation $\mathbb{E}$ refer in the formula before (5)?
>
> **A:** Thanks. This expectation is calculated over all $Y_t\sim Q_t$ and all $Y_{t_k}\sim Q_{t_k}$. We has duly added this explanation.
>
> Reference:
>
> [1] Joe Benton, Valentin De Bortoli, Arnaud Doucet, et al. Linear convergence bounds for diffusion models via stochastic localization. International Conference on Learning Representations. 2024.
>
> [2] Holden Lee, Jianfeng Lu, and Yixin Tan. Convergence for score-based generative modeling with polynomial complexity. In NeurIPS. 2022.
>
> [3] Santosh Vempala and Andre Wibisono. Rapid convergence of the unadjusted langevin algorithm: Isoperimetry suffices. In NeurIPS. 2019.
>
> [4] Sitan Chen, Sinho Chewi, Jerry Li, et al. Sampling is as easy as learning the score: theory for diffusion models with minimal data assumptions. International Conference on Learning Representations. 2023.
>
> [5] Hongrui Chen, Holden Lee, and Jianfeng Lu. Improved Analysis of Score-based Generative Modeling: User-Friendly Bounds under Minimal Smoothness Assumptions. In International Conference on Machine Learning. 2023.
>
> [6] Shivam Gupta, Aditya Parulekar, Eric Price, et al. Sample-efficient training for diffusion. 2023.
>
> [7] Javier E. Santos, Yen Ting Lin. Using Ornstein-Uhlenbeck Process to understand Denoising Diffusion Probabilistic Model and its Noise Schedules. 2023.

---

### Official Review · Reviewer_AwUr · 2024-03-21

**Q2-1 Originality-Novelty:** 3
**Q2-2 Correctness-Technical Quality:** 3
**Q2-5 Clarity Of Writing:** 4

**Q1 Summary And Contributions:**

This paper studies the stepping schedule tuning in diffusion models. Motivated by the drawbacks of the current fixed stepping schedule, which include the lack of theoretical foundations and assurance of optimal performance at chosen discretization points, this paper proposes adaptive time-stepping schedules and designs algorithms based on it for both continuous diffusion and discrete diffusion. Experiments show that the new proposed two algorithms manage to generate great samples with shaking score errors, and improve generating performance for both well-trained and under-trained models.

**Q2-3 Extent To Which Claims Are Supported By Evidence:**

3: Good: the main claims are supported by convincing evidence (in the form of adequate experimental evaluation, proofs, (pseudo-)code, references, assumptions).

**Q2-4 Reproducibility:**

3: Good: key resources (e.g. proofs, code, data) are available and key details (e.g. proofs, experimental setup) are sufficiently well-described for competent researchers to confidently reproduce the main results.

**Q3 Main Strengths:**

1. The paper is well-written and highly readable.
2. The main idea of this paper is novel. The paper clearly points out the weakness of the current fixed stepping schedule, and the two proposed algorithms solve the problems to some extent.
3. The two proposed algorithms have solid theoretical foundations.
4. The proposed algorithms achieve better empirical performance compared to the current fixed stepping schedules.

**Q4 Main Weakness:**

1. The empirical performance is not that strong especially when the number of discretization points are large. And also, the performance of the newly proposed GC shows randomness in iteration numbers, i.e the performance isn't get better when there're more iterations, and sometimes with 5 iterations the performance of GC is even worse than fixed-stepping schedule. If authors can explain about the reasons of this randomness, then the paper will sound more strong.
2. The idea of two algorithms are quite great, but since the process of adjusting discretization points is optimizing one by one, it's very likely that the final results get stuck in a local optima, especially when the choice of the first several points is not that good.
3. A small suggestion is that the figures in the paper can be adjusted to be more beautiful. For example, in Figure 2 (Left), Figure 3 (Left), the labels on the left of the figure overlap with the lines and should be moved to the right.

**Q5 Detailed Comments To The Authors:**

Please see the weakness above.

**Q9 Complying With Reviewing Instructions:**

Yes

---

> ### Author Rebuttal · Authors · 2024-04-05
>
> Thank you for your constructive comments and kind support! All your concerns have been carefully addressed as below. The manuscript is carefully revised accordingly. We sincerely hope our responses fully address your questions.
>
> **Q1:** The empirical performance is not that strong especially when the number of discretization points are large. And also, the performance of the newly proposed GC shows randomness in iteration numbers, i.e the performance isn't get better when there're more iterations, and sometimes with 5 iterations the performance of GC is even worse than fixed-stepping schedule.
>
> **A:** Thanks. We provide some explanation for this phenomenon.
> - KL/TV in theory with FID in experiments. We focus on optimizing the upper bounds of KL and TV but FID is used as evaluation score in experiments. We cannot use KL and TV in experiments because target distributions are unknown.
> - Convergence bounds may be not tight. There may be gap between the inner sampling error and convergence bounds, so do EB. This makes optimizing EB different from directly minimizing sampling error.
> - DAE is not tight for score matching error. The difference between DAE and the score matching error is large when $t$ is small, and small when the $t$ is high. This may cause our algorithm to select too few points with small noise levels, leading to suboptimal results.
>
> **Q2:** Since the process of adjusting discretization points is optimizing one by one, it's very likely that the final results get stuck in a local optima,
>
> **A:** Thanks. The existence of local minima is a common problem in deep learning, and also in diffusion models, but our experiments show that even if our algorithm finds a local minimum, our method gains significant performance improvement.
>
> **Q3:** Minor issues in presentation: the labels on the left of the figure overlap with the lines and should be moved to the right.
>
> **A:** Thanks and addressed.

---

### Meta-Review · Area_Chair_ZhVp · 2024-04-18

This paper proposes a novel method for diffusion models. The key idea is to adaptively adjust the time steps used in the diffusion process to improve the quality of the generated data. The authors propose two algorithms: one for continuous diffusion and one for discrete diffusion. They show that their methods improve the generation ability of diffusion models on standard benchmarks.

Strengths:

Introduces a novel approach to improve diffusion models by adaptively adjusting time steps.
Shows improved generation ability on MNIST, CIFAR-10, and CELEBA benchmarks.
Provides theoretical foundation for the proposed algorithms.

Weaknesses:

Clarity and presentation: The paper is unclear and difficult to understand in some parts. Notation is not always well-defined, and some figures are unreadable.
Lack of strong benchmarks: The authors only evaluate their method on relatively simple datasets (MNIST, CIFAR-10) that are not commonly used for modern generative models.
Missing proofs: Some reviewers requested proofs for key theorems but did not receive them from the authors.